# Microencapsulation of Analgesics as an Analog Form of Medicine

**DOI:** 10.3390/pharmaceutics17070916

**Published:** 2025-07-15

**Authors:** Aidana Nakipekova, Bates Kudaibergenova, Arkady S. Abdurashitov, Gleb B. Sukhorukov

**Affiliations:** 1Faculty of Chemistry and Chemical Technology, Al-Farabi Kazakh National University, 71 Al-Farabi Ave., Almaty 050042, Kazakhstan; aidananakipekova@gmail.com; 2Vladimir Zelman Center for Neurobiology and Brain Restoration, Skoltech, 3 Nobel Str., Moscow 121205, Russia; a.abdurashitov@lift.center; 3Life Improvement by Future Technology (LIFT) Center, Bolshoi Bl. 30, Moscow 121205, Russia

**Keywords:** microencapsulation, water-soluble drug, prolongation effects, biodegradable polymers

## Abstract

**Objectives:** This research focuses on the development of fabrication approaches for microparticles intended for controlled drug delivery. The primary objective is to identify the most suitable polymer type, particle size, and morphology for encapsulating a water-soluble crystalline drug. Optimizing these parameters may enhance structural stability and prolong the release of this active substance. **Methods:** The microparticles were fabricated through the encapsulation of a drug substance within a polymer carrier and employing polymer casting on prepatterned surfaces, followed by the loading of drug precipitates and the application of a sealing layer. The crystalline powder 1-allyl-2,5-dimethylpiperidol-4 hydrochloride served as the core cargo material, while the walls of these particles were composed of polylactic acid (PLA) and a poly (α-caprolactone) (PCL) in a 70:30 composition ratio. **Results:** The size and volume of the microparticles were found to be dependent on the geometric parameters of the template and the concentration of the polymer solutions. The study demonstrates the formation, physical dimensions, and particle count at varied polymer compositions and concentrations. The formation of the PLA and PCL mixture occurred solely through physical interactions. Scanning electron microscopy (SEM) and optical microscopy were employed to observe the appearance and physical dimensions of the microparticles. The obtained data confirm that tailored polymer compositions can yield consistent particle morphology and a suitable drug elution rate. **Conclusions:** The results indicate that microparticles sealed with an optimal polymer composition exhibit enhanced release properties. This finding highlights the feasibility of microencapsulation at precise ratios and concentrations of polymers to achieve the long-lasting effects of water-soluble drugs.

## 1. Introduction

Microencapsulation is a cutting-edge technique that has gained significant attention across various scientific disciplines. This innovative process involves enclosing tiny drug particles or droplets within a protective polymer coating, resulting in microcapsules with unique properties.

In terms of controlling release, microencapsulation enables precise regulation over the release of active substances, such as drugs or nutrients entrapped within the microcapsules. Optimization of the composition and shell thickness leads to designing controlled-release systems that can retain and deliver substances at specific rates, ensuring prolonged or targeted effects. Micro- and nanoencapsulation provide protection, opportunities for external triggering, improved efficacy and therapeutic benefits, and provides bioavailability, especially for poorly water-soluble drugs [1,2,3,4,5,6,7,8,9,10,11].

Nanotechnology-based microencapsulation facilitates the fabrication of stimuli-responsive systems that are tuned to specific triggers, such as temperature, pH, enzymes, or light, to release the encapsulated drug at the desired site. By incorporating stimuli-responsive polymers or coatings, scientists can achieve targeted drug delivery, minimizing side effects and optimizing therapeutic efficacy.

There is a growing emphasis on using biodegradable and biocompatible polymers in pharmaceutical microencapsulation. Polymers such as poly (lactic-co-glycolic acid) (PLGA), chitosan, and gelatin are widely employed due to their excellent biocompatibility and controllable degradation properties. These materials offer the advantage of sustained drug release and eliminate the need for surgical removal. Biodegradable microcapsules also minimize toxicity and immunogenicity concerns. Ongoing research focuses on optimizing the formulation parameters and exploring novel biodegradable materials to meet specific drug delivery requirements.

Three-dimensional printing technologies and additive manufacturing provide the precise fabrication of complex structures, enabling the production of customized microcapsules with tailored properties; 3D-printed microcapsules offer precise control over size, shape, and release kinetics, facilitating personalized medicine and patient-specific dosing. The integration of additive manufacturing and 3D printing with microencapsulation techniques paves the way for on-demand manufacturing and personalized drug delivery systems [12,13,14,15,16,17,18,19,20].

The primary role of analgesic drugs is to manage pain effectively. They target various mechanisms involved in pain transmission, perception, and modulation, providing relief across a wide range of conditions, including acute and chronic pain, postoperative pain, neuropathic pain, and inflammatory pain. Analgesic drugs act by blocking pain signals at the site of injury or within the central nervous system (CNS), altering neurotransmitter release, or modulating pain receptors, resulting in reduced pain intensity and improved patient comfort.

The increasing demand for analogs of analgesic drugs stems from the need for personalized medicine, enhanced safety and efficacy, novel drug delivery systems, and addressing the opioid crisis. By developing analogs with tailored properties and improved pharmacokinetics, researchers aim to optimize pain relief while minimizing side effects and customizing treatment to individual patients. Furthermore, the exploration of novel drug delivery systems and the development of alternatives to opioids contribute to the expanding demand for analogs in various applications.

Continued research and innovation in the field of analgesic drugs and their analogs hold the potential to revolutionize pain management and improve patient outcomes. By further understanding pain pathways and developing targeted therapies, healthcare professionals can provide more precise and effective pain relief, meeting the diverse needs of patients across different conditions and pain types [21,22].

In our work, we elaborate approaches to encapsulating analgesic drugs such as richlocaine into microcapsules made of biodegradable polymers, exploring templates of different shapes. We utilized the drug richlocaine, which possesses analgesic properties and dermatoprotective activity. It is superior in the breadth of therapeutic action to many reference drugs of this group, has low toxicity, does not cause toxic and allergic reactions, and is well tolerated by patients.

## 2. Materials and Methods

Polycaprolactone (PCL) with a molecular weight (Mw) of approximately 80,000 and polylactic acid (PLA) with a molecular weight of approximately 60,000 were obtained from Sigma-Aldrich, Schnelldorf, Germany. The analgesic drug with dermatoprotective activity, richlocaine, is a white or slightly yellowish crystalline powder that is odorless, with the chemical formula 1-allyl-2,5-dimethylpiperidol-4 hydrochloride. Richlocaine is highly soluble in water and isotonic sodium chloride solution.

### 2.1. Polymer Composite Preparation

To prepare the polymer composite, solutions of the polymers were initially prepared separately. Dissolving polymers to obtain initial working solutions for mixing into the composite is a crucial step in the technological process. A predetermined amount of solvent is used to achieve the desired concentration, and the mixture is stirred on a magnetic stirrer for one hour, with the container tightly sealed.

The polymer composite preparation was carried out using a laboratory magnetic stirrer—IKA C-MAG 7 digital, manufactured by IKA (Königswinter, Germany). The polymers were dissolved in dichloromethane in tightly sealed borosilicate glassware to prevent solvent evaporation. To fabricate a composite polymer, individual solutions were initially prepared separately due to differing solubility temperatures. Solutions with concentrations of 1%, 1.5%, and 3% were formulated. Following preparation, the solutions were combined in a volumetric ratio of 30:70 based on mass, ensuring uniform distribution within the composite matrix, and stirred at room temperature for around 50–60 min at 200–250 rpm.

### 2.2. Fabrication of Sealed Free-Standing Polymer Films with Microchamber Array

A microchamber array film was used, consisting of a patterned film with microwells, a drug, and a flat film. The patterned film was formed using a prefabricated PDMS stamp with wells [23,24]. The stamp dimensions were 8 × 8 mm, with the following specifications for the microwells: microwell #1 had a diameter of 30 μm, a depth of 15 μm, and a center-to-center distance of 40 μm; microwell #2 measured 300 μm in length and 50 μm in width, had a depth of 25 μm, and had a center-to-center distance of 150 μm; microwell #3 had a length of 70 μm, a width of 50 μm, a depth of 25 μm, and a center-to-center distance of 150 μm. Specifically, the prefabricated mold—containing predefined microchamber geometry—was immersed in a pre-prepared 1 wt%, 3 wt% PCL, PLA, or PCL/PLA composite chloroform solution for 5 s and then removed and suspended vertically to dry under ambient conditions. During drying, the solvent gradually evaporates, resulting in the formation of a thin polymer film forming on the PDMS mold’s topography. This initial film serves as the outer shell of the future microcapsules.

After complete drying, the active pharmaceutical ingredient was loaded into the microchambers, followed by the application of a prefabricated flat polymer film as a sealing layer. This top film was gently laminated over the filled surface to enclose the contents and form sealed microcapsules. The combined structure was allowed to settle, ensuring proper adhesion and sealing.

### 2.3. Manufacturing of PDMS Molds

PDMS molds were created using a laser pyrolysis approach [25]. First, two components of PDMS were thoroughly mixed in a manufacturer-prescribed ratio of 10:1. When PDMS was poured into a rectangular mold (width—22 mm; height—50 mm, depth—4 mm) and degassed under vacuum (<0.3 Bar) for 30 min to remove air bubbles. After this step, the mold was placed in an oven at 60 C for 4 h. Completely cured PDMS was removed from the mold and manually coated with pyrolysis initiator—soot carbon nanoparticles. Focused laser light was used to ignite the pyrolysis process. Irradiation of the DPSS laser (100 mW, 532 nm) was collected and focused onto the coated surface of the PDMS using a microscope objective lens (NA 0.2, estimated focus spot ~3.2 µm). Two high-precision translation stages (STANDA: 8MT167M-25BS1 and 8MTL1301-170-LEn1-100, Vilnius, Lithuania) and a self-written software were utilized to control the size and quantity of microwells. We explored three different laser operation regimes to produce different microwells. These are the following and can be seen on Figure 1:

Microwell #1: Laser power—100 mW, laser pulse length—5 ms, number of pulses—5, inter-pulse delay—10 ms.

Microwell #2: Laser power—100 mW, sample linear velocity—2.5 mm/s, laser on time—100 ms, laser off time—40 ms.

Microwell #3: Laser power—100 mW, sample linear velocity—2.5 mm/s, laser on time—24 ms, laser off time—10 ms.

A promising approach to manufacturing carriers of a specified shape is based on creating templates on a consumable substrate using appropriate lithographic techniques, their templating, subsequent shell material deposition, cargo loading, and sealing [26,27,28]. Several loading methods were employed for encapsulating richlocaine within a polymeric shell.

Method #1 involved the capillary-driven flow of liquid polymer along the walls of a mold. To achieve this, a stamp was immersed in a polymer solution and left to dry on a heated surface at 60–90°, allowing the polymer to flow along the mold wall and trimming the edges along the template protrusion. Subsequently, the active ingredient (drug) was loaded, followed by the application of a polymer film covering. After coating, the assembly was placed back onto the heated surface. The polymer melted and flowed into the well, thus trimming the microcapsule edges and sealing them. The microcapsules were extracted from the mold using a gelatin solution onto a flat surface and cooled to allow gelation. The gelatin layer adheres to the capsules, enabling their separation from the mold. Subsequently, the gelatin can be removed by gentle water washing and decantation.

Method #2 was based on sealing the drug between polymer layers. This involved raising the temperature at the interface of the polymer films under a certain load, causing them to melt, fuse together, and cut through. As a result, sealed microcapsules were obtained within the wells on the stamp. These were then transferred onto a polyvinylpyrrolidone film as a receiving surface. The mold is gently pressed onto the film, transferring the microcapsules without exposing them to moisture. This dry transfer method preserves capsule integrity and allows storage in solid form.

### 2.4. Characterization Technique

The morphology and size distribution were assessed using scanning electron microscopy (SEM) and optical microscopy. Additionally, this drug delivery system was evaluated by determining capsule capacity and drug release. Scanning electron microscopy (SEM) measurements were performed with a VEGAIII (TESCAN, Brno, Czech Republic) microscope at an operating voltage of 30 kV. Before measurement, gold was sprayed onto the sample (approximately 5 nm gold layer) using an Emitech K350 sputter-coater (Quorum Technologies Ltd., Ashford, UK).

The release was carried out in the physiological solution (0.9% NaCl in deionized water) at 37 °C. To ensure uniform mixing, vials were subjected to constant stirring on an orbital shaker at 300 RPM, then 5 mcg of polymeric particles loaded with richlocaine were added to 1 mL of saline solution. To carefully study the elution rate at its early moments, the first samples were taken at 4, 8, and 24 h, respectively. The rest of the data points were sampled every 24 h until reaching the 72 h mark. At each time point, the Eppendorf (Hamburg, Germany) vials were centrifuged to separate particles from the solution. The whole supernatant was then taken for spectroscopic analysis, and particles were resuspended in the fresh, clean portion of the physiological solution. The spectroscopic study was carried out as follows: 200 ul of supernatant was placed in the UV-transparent 96-well plate, and absorption at 275 nm was evaluated. Using the equation of the calibration curve, the absorbance value was converted to the richlocaine concentration. The release ability was assessed using a dual-mode microplate reader with monochromator-based optics for absorbance and sensitive top- and bottom-reading fluorescence applications, specifically the Tecan Infinite M Nano+, Männedorf, Switzerland.

IR spectroscopy was performed on a Nicolet iS10 FT-IR Spectrometer (ThermoFisher Scientific, Waltham, MA, USA).

## 3. Results and Discussion

The reason for exploring richlocaine, in addition to its local anesthetic activity, is that it exhibits pronounced antiarrhythmic, analgesic, anticonvulsant, hepatoprotective, and antimicrobial effects. New dosage forms of richlocaine with a long-lasting action and pharmacological compositions that include energy-providing and antioxidant substances have been developed. The study of richlocaine for new indications and its various pharmacological compositions is at different stages of preclinical and clinical trials [29].

Poly (lactic acid) (PLA) is a biodegradable and biocompatible polymer that finds applications in packaging and medicine. Its precursor is lactic acid, making PLA an environmentally friendly material derived from renewable resources. Currently, the main focus of research lies in the development of simple and economically efficient manufacturing methods suitable for scaling up and transitioning to the medical market, aiming to broaden the utilization of PLA within the biomedical industry [30,31,32].

The second polymer selected for microcapsule fabrication was polycaprolactone PCL. The use of PCL with a higher molecular weight for encapsulation resulted in the formation of core-shell microcapsules with a smoother surface and slightly reduced average shell thickness. Additionally, an enhanced shell stability against air moisture diffusion over an extended storage period was observed for these microcapsules [33].

In addition to several advantages, polylactic acid (PLA) has drawbacks such as brittleness and relatively high glass transition and melting temperatures. However, copolymerizing PLA with other polymers enhances its characteristics, allowing for the creation of a desired material with preferred physical properties [32,34].

Based on the obtained microcapsules and the investigation of their physical properties, a decision was made to utilize a composite of these polymers, specifically leveraging the strength of PLA and the low melting point and elasticity of polycaprolactone (PCL) [35]. The polymer ratios employed to produce the microcapsules are presented.

Solutions of 1%wt and 3%wt PLA and PCL polymer in chloroform were utilized. To be avoided, if possible, there are some procedures that reduce their remaining quantities in final particles [36,37]. According to toxicity data, a relatively small dosage of particles (<10 ug/mL) exhibits minimal cytotoxic effect, but higher dosages clearly show signs of toxicity, which can be related to the residual chloroform content, as well as several other factors. The main goal of this article was to highlight the novel dosage form of the richlocaine, suitable for prolonged therapy. In principle, in a case of pure PLA, chloroform can be avoided and replaced with a “green” solvent like ethylacetate. The choice of polymer concentration was governed by the microcapsule shape. For well-shaped microcapsules, the 3% solution proved excessively viscous, filling the wells without creating space for loading the active ingredient (richlocaine). Conversely, the film formed from a 1% polymer solution for elongated microcapsule shapes was too thin, resulting in inadequate sealing adhesion. Furthermore, due to the polymer’s viscosity, microcapsules were extracted from the stamp with difficulty, with lateral stretching and rupture observed. Thus, it is concluded that smaller capsule volumes require a lower percentage of polymer solution, while larger microcapsule shapes necessitate a higher percentage for enhanced structural integrity.

In addition to optimizing drug loading, the incorporation of pre-ground richlocaine crystals dispersed in alcohol enhanced the homogeneity of the drug distribution within the microcapsules, ensuring a creamy consistency conducive to effective loading. This meticulous preparation strategy yielded a notable drug loading efficiency exceeding 70%. In stark contrast, the dry loading method, wherein the drug was introduced directly into microwells without the pre-dispersion process, exhibited non-uniform filling, achieving a mere 30% of the well capacity.

Exploring the physical attributes of the microcapsules, not completely spherical configurations were considered for their inherent advantages [38,39,40,41,42,43,44]. These non-spherical shapes, namely Form 1 (well-shaped), Form 2 (elongated boat-shaped), and Form 3 (semi-oval-shaped), provide improved flow characteristics and higher packing capacities compared to their spherical counterparts. The choice of half-sphere wells (Form 1) for stamping microcapsules combines practical considerations such as volume optimization, uniformity, material efficiency, mechanical stability, and packing efficiency, making it a suitable geometric form for this application. Boat-shaped Form 2 provides a maximal volume for the microcapsules within a given footprint, maximizing the payload capacity of each capsule. The semi-oval-shaped Form 3 offers inherent mechanical stability, reducing the risk of deformation or collapse during manufacturing and handling in comparison to Form 2.

The curved surface of all three Forms of wells promotes the efficient packing of microcapsules, minimizing void spaces and maximizing storage density. The determination of concentration in the solution facilitated the establishment of individual microcapsule capacities, with Form 1 exhibiting a capacity of 3 ng, Form 2 with 30 ng, and Form 3 with 12 ng (Figure 2).

To delve further into the physics of these microcapsule shapes, the non-spherical geometries contribute to enhanced packing efficiency due to reduced interparticle void spaces and improved interlocking capabilities. The elongated boat-shaped Form 2, for instance, offers an increased surface area for drug loading and potential sustained-release benefits. Additionally, the semi-oval-shaped Form 3, with its unique curvature, presents opportunities for controlled drug release profiles through the manipulation of surface area and volume ratios. These nuanced physical considerations underscore the potential impact of microcapsule geometry on drug loading and overall performance.

Based on the data obtained, it is possible to determine the potential methods of microencapsulation. The proposed principle is illustrated in Figure 3.

First Microcapsule Formation Method: After loading the active ingredient richlocaine, the stamp was left on a heated surface to melt the top polymer layer for sealing. As observed in the scanning electron microscopy (SEM) images, the lower polymer layer underwent melting, causing the drug to transition into the liquid polymer phase due to gravity and capillary-driven flow. This liquid polymer phase was washed away upon extraction from the stamp, as evidenced by the micropores on the microcapsule walls. This method has the potential to produce microporous particles using other salts as carriers or adsorbents, which could enter the cavity of the microcapsule through the existing pores.

Additionally, there were issues with polymer adhesion between layers; as the lower polymer layer dried, it contracted, hindering the loading of the richlocaine. While the method is feasible, the productivity per batch is less than 35–40%, which renders it unprofitable. However, this method is not suitable for polylactic acid due to its higher melting temperature of 180 °C, which significantly exceeds the melting temperature of richlocaine at 130 °C.

The outcomes of microcapsule fabrication are visually depicted in the scanning electron microscopy (SEM) images presented below (Figure 3). The experimentation involved the utilization of two distinct templates, aiming to elucidate the influence of microcapsule geometry on their extraction from production molds. As evident from the depicted images, the structural integrity of the microcapsules is compromised. Specifically, one instance reveals inadequate polymer coverage on the semi-spherical form of the capsule, while another highlights the inadvertent release of medicinal substance crystals during the extraction process from the manufacturing molds. Presumably, micropores form on the microcapsule shell due to the flushing of the encapsulated drug during extraction processes. The method involved heating a template on a hot plate to facilitate polymer flow into the mold, with the subsequent loading of richlocaine. Since the polymer did not have time to cool, richlocaine was immersed into the molten polymer, followed by solidification. During extraction, richlocaine leached into the solution, creating voids within the polymer matrix and resulting in the formation of micropores on the polymer shell (Figure 4).

Second Microcapsule Formation Method: A film with a thickness of approximately 200 microns was individually fabricated using 1%, 1.5%, and 3% solutions of PLA and PCL polymer solutions on a polypropylene substrate. The process involved immersing a stamp in the polymer solution, allowing it to dry, and subsequently loading the microwell with the drug. The loaded microwell was covered with a prefabricated film and subjected to heat pressing for 5 min. The press temperature was incrementally raised from 60 °C to 75 °C, accompanied by an increase in pressure from 5 to 6 kg per cm^2^. Following the cooling phase, the trimmed polymer mesh layer was meticulously removed from the inter-row space.

The microcapsules were then transferred from the stamp into a saline solution, and the release of richlocaine was assessed. Alternatively, for long-term storage, consideration can be given to depositing the microcapsules onto a polyvinylpyrrolidone film. This approach facilitates the storage of microcapsules in a dry film form, which readily dissolves in water upon use.

The outcomes are elucidated through the subsequent scanning electron microscopy (SEM) depictions (Figure 5). The experimental procedure employed three distinct templates to clarify how microcapsule geometry influences their extraction from production molds. Notably, all instances reveal well-sealed capsule edges and an intact shell structure, ensuring the preservation of the internal capsule contents. The surface roughness of the capsule shell is attributed to the imprint of the template surface during shell formation, a characteristic that is non-essential for the primary objective.

Upon thermal pressing, capsule edges are hermetically sealed, leading to a reduction in the inter-row space thickness. Consequently, during the template removal, microcapsules undergo separation, as evident in the figures by discernible tear marks along the capsule edges. This phenomenon is a consequence of the thinning of the inter-row space and demonstrates the efficacy of the method in achieving distinct microcapsule entities.

In order to validate the volumetric completeness of the microcapsules, cross-sectional images were captured, revealing clear visibility of their internal contents (Figure 6). The assessment of uniform medicinal substance density remains a challenge, primarily attributable to the manual application process. Future refinements in this manual procedure are anticipated to enhance homogeneity. The mechanical properties of the capsules, characterized by notable flexibility, preclude brittleness or excessive softness that could compromise their structural integrity and potentially impact drug release kinetics.

In the realm of physico-mechanical considerations, it is noteworthy that the flexibility of the capsules contributes to their resilience during handling and prevents deformations that may impede the controlled release of the encapsulated drug. Additionally, an evaluation of the capsules’ elastic modulus and tensile strength could provide valuable insights into their mechanical stability, warranting further exploration in future investigations.

Considering the SEM images acquired, an assessment of the release kinetics was conducted for a cohort of microcapsules characterized by a shell composition comprising PLA (polylactic acid) and PCL (polycaprolactone). The preference for this batch stemmed from its superior performance and sealing attributes, surpassing alternative polymer ratios considered for microcapsule shell formation.

Fourier transform infrared spectroscopy (FTIR) was employed to confirm the presence of the active substance in the microcapsules (Figure 7). The analysis was conducted using a Nicolet iS10 FT-IR (ThermoFisher Scientific, Waltham, MA, USA) instrument equipped with an ATR attachment, which eliminated the need for complex sample preparation. Microcapsule samples were applied directly to the ATR crystal. The resulting spectrum exhibited characteristic absorption bands at 1700 cm^−1^ (C=O stretching), 3400 cm^−1^ (N–H or O–H stretching), 1600 cm^−1^ (aromatic ring), and 1250 cm^−1^ (C–N or C–O bonds), thereby confirming the presence of richlocaine in the polymer matrix of the microcontainers.

Delving into the physics underlying the observed performance, it is imperative to acknowledge the interplay of polymer composition in influencing shell properties. The enhanced outcomes observed in the PLA and PCL microcapsules can be attributed to their synergistic mechanical and thermal characteristics, resulting in a more resilient and impermeable encapsulation matrix. Further investigations into the mechanical modulus and thermal conductivity of these specific polymer blends are warranted to elucidate the physico-mechanical principles governing their superior performance. The liberation kinetics of richlocaine from microcapsules were scrutinized within a saline solution using a thermostatic shaker. Upon careful examination of the release profile, it becomes evident that a predominant fraction of the drug is liberated within the initial 8 h period, with sustained release plateauing thereafter, exhibiting minimal changes even at the 24 h mark (Figure 8). It is common to use the Korsmeyer–Peppas Model [45] for describing drug elution from polymeric microstructures. It is clear from the release curve that the elution is non-linear and could be approximated by the power law to a certain accuracy. Korsmeyer–Peppas fitting reveals the elution rate constant value of 0.7522 and the power value of 0.092, which indicates purely diffusional drug release, potentially caused by the highly porous wall of particles.

Additionally, we do not exclude the possibility that the observed high initial release may be partially attributed to structural defects in the microcapsules, such as incomplete sealing or thin regions in the outer polymer layer, leading to premature leakage.

Continuing our investigation, optical microscopy reveals internal compaction within the microcapsules. This observation suggests that the microcapsules, while maintaining structural integrity and volumetric completeness, exhibit diminutive sizes, precluding a definitive determination of their content through current optical microscopy techniques. Further explorations, possibly involving advanced imaging modalities, are imperative to unravel the intricacies of the microcapsule morphology and its correlation with drug release kinetics.

Although the particle’s stability is an important factor for long-term drug elution (weeks to months), experimental data confirms that the described dosage form of richlocaine is suitable for sustained dosage within the first 24 h. At such timescales, both PLA and PCL are highly stable [46], and changes in particles due to hydrolysis and enzymatic activities can be neglected.

Additional in vitro experiments were conducted to evaluate the cytotoxicity and biological activity of the synthesized microparticles. One of the key analytical methods employed was the IC50 assay, which quantitatively determines the concentration required to inhibit 50% of cellular activity within the target culture. The IC50 method was applied to Leishmania major and Leishmania tropica cell lines after a 72 h incubation at 23 °C. The microparticle concentrations ranged from 1 to 60 μg/mL, and optical density measurements were recorded spectrophotometrically at 570 nm.

At low concentrations (1–10 μg/mL), the microparticles exhibited minimal cytotoxicity, maintaining cell viability above 85%. At 60 μg/mL, viability decreased to approximately 60%. Higher doses (>60 μg/mL) resulted in a significant reduction in cell viability (<30%), indicating a dose-dependent toxicity profile.

The initial microparticle dimensions (50 × 50 × 15 μm) were increased fivefold to 250 × 50 × 60 μm to enhance drug-loading capacity. However, at high volume ratios, these enlarged microparticles led to the localized accumulation of richlocaine, exacerbating toxicity. Consequently, size optimization was implemented, reducing microparticle dimensions to 120 × 60 × 60 μm. This modification ensured a balanced approach, maintaining efficient drug encapsulation while enhancing biocompatibility and supporting stable active compound release without observable toxicity.

The IC50 analysis enabled the establishment of a safety threshold for the application of richlocaine-loaded microparticles. The results confirmed that particle size plays a critical role in determining cytotoxic effects. The optimization of microparticle dimensions (to 120 × 60 × 60 μm) facilitated high therapeutic efficacy while preserving biocompatibility.

Although the presence of the active pharmaceutical ingredient (richlocaine) in the polymer microcapsules was confirmed by Fourier transform infrared spectroscopy (FTIR), which provided a reliable qualitative assessment, further quantitative analysis is essential to fully characterize the system’s performance. In future studies, high-performance liquid chromatography (HPLC) will be implemented to determine encapsulation efficiency, drug loading capacity, and precise release kinetics over time under simulated physiological conditions. This approach will allow for a more rigorous evaluation of the delivery system’s therapeutic potential and will enable a direct comparison with existing microencapsulation technologies.

## 4. Conclusions

This study systematically evaluated two fabrication approaches for microcapsule formation using identical polymer compositions. The first method—employing thermal sealing with a patterned stamp—demonstrated limited scalability due to low per-batch yield and suboptimal interlayer adhesion, particularly when utilizing polylactic acid (PLA), whose elevated melting point posed processing challenges.

In contrast, the second method, involving the solution casting of PLA and polycaprolactone (PCL) to form a sealing film, exhibited improved encapsulation quality and operational consistency. Scanning electron microscopy (SEM) confirmed the formation of well-sealed microcapsules with favorable surface morphology, while physico-mechanical analysis underscored the complementary thermal and mechanical properties of the PLA/PCL matrix, contributing to the capsules’ stability and impermeability. Kinetic studies of drug release revealed an initial rapid release phase within the first 8 h, followed by a prolonged plateau, indicating effective controlled-release behavior.

Collectively, these findings underscore the critical role of polymer composition and process optimization in achieving reproducible particle morphology and controlled release kinetics. The study reinforces the relevance of PLA and PCL-based microcapsules in drug delivery, especially for water-soluble crystalline drugs, and provides a foundation for future work targeting improvements in polymer selection, fabrication scalability, and analytical characterization.

Although significant progress has been made in the development of biodegradable microparticles for sustained drug delivery, challenges persist, particularly in the cost-effectiveness of biopolymers and the establishment of robust evaluation methods for bioadhesive platforms. Continued interdisciplinary research is essential to bridge technological and biological gaps and to advance safe, efficient, and patient-tailored drug delivery systems.

## Figures and Tables

**Figure 1 pharmaceutics-17-00916-f001:**
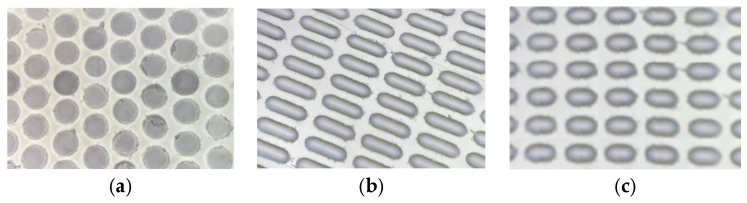
Microwell matrixes on an optical microscope: (**a**) microwell #1, (**b**) microwell #2, (**c**) microwell #3.

**Figure 2 pharmaceutics-17-00916-f002:**
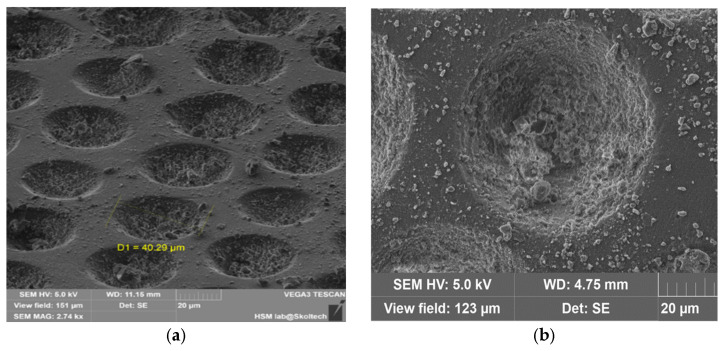
Filling wells with drug crystals: (**a**) general view; (**b**) solitary view.

**Figure 3 pharmaceutics-17-00916-f003:**
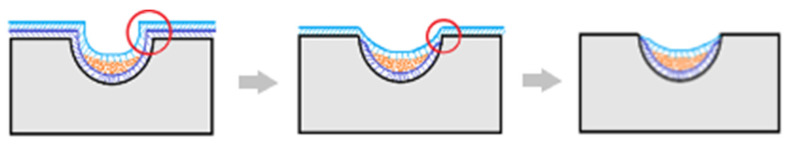
Principle of microencapsulation: purple and blue—polymer layers, orange dots—drug. Red circle indicates the template edge, on which the separation to the individual particles occurs upon heating.

**Figure 4 pharmaceutics-17-00916-f004:**
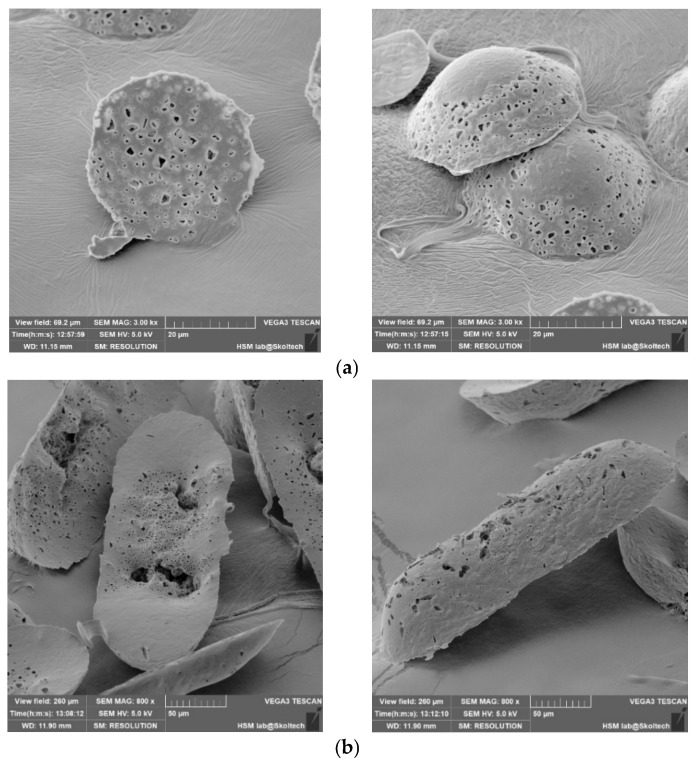
Microcapsules by Method #1: (**a**) microwell #1, (**b**) microwell #3.

**Figure 5 pharmaceutics-17-00916-f005:**
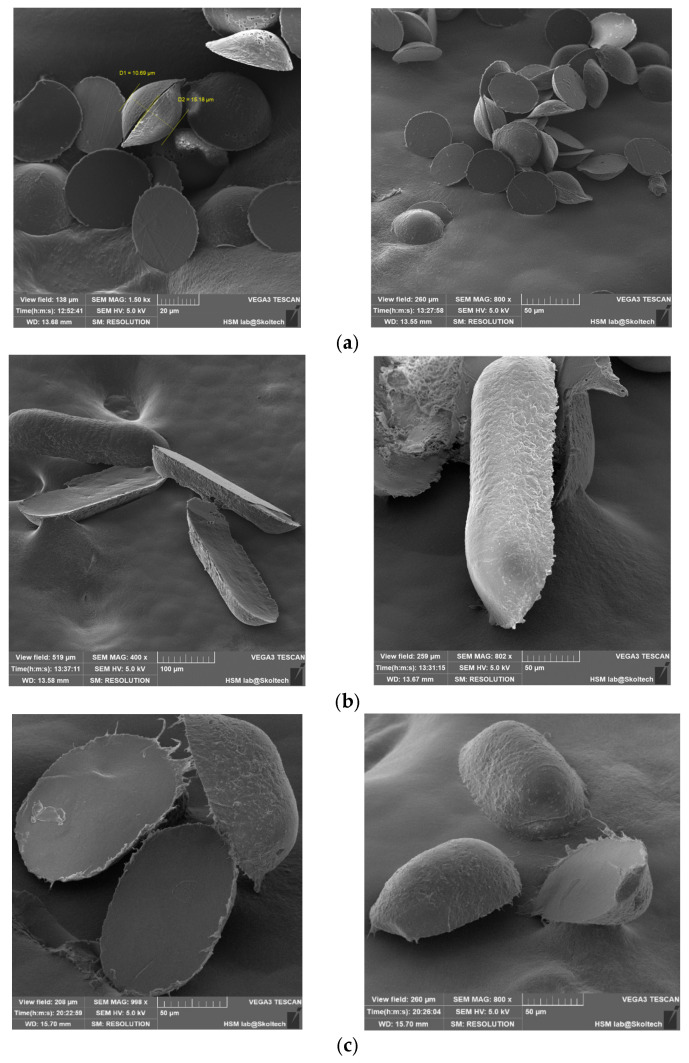
Microcapsules by Method #2: (**a**) microwell #1, (**b**) microwell #2, (**c**) microwell #3.

**Figure 6 pharmaceutics-17-00916-f006:**
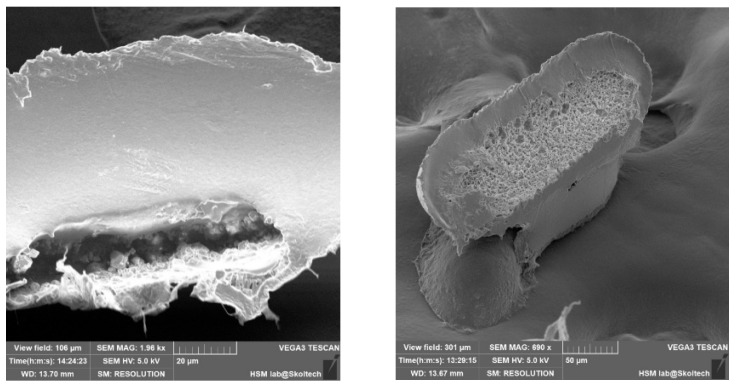
Internal contents of microcapsules.

**Figure 7 pharmaceutics-17-00916-f007:**
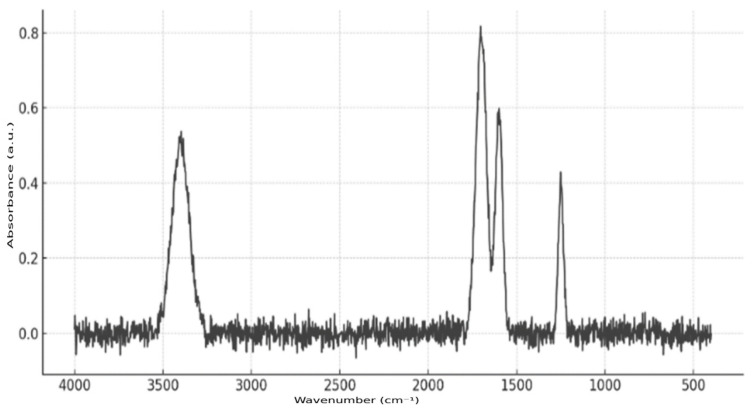
Richlocaine FTIR spectrum.

**Figure 8 pharmaceutics-17-00916-f008:**
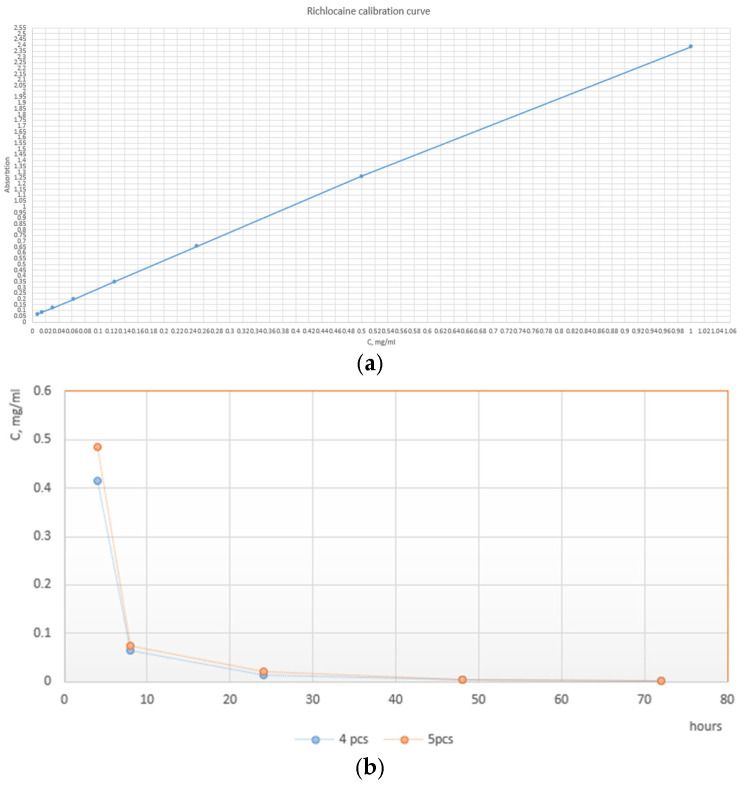
(**a**) Richlocaine calibration curve; (**b**) release of drug from PCL-PLA microcapsule.

## Data Availability

Data is contained within the article.

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
