# Peer review of "Microencapsulation of Analgesics as an Analog Form of Medicine"

_pharmaceutics, 2025, doi:10.3390/pharmaceutics17070916_

Round 1
Reviewer 1 Report
Comments and Suggestions for Authors
This article reports a method for preparing microcapsules based on molds. However, before publication, there are several issues that need to be addressed.
1.The microcapsules are oval in shape. How can we ensure the uniformity of the outer shell of the microcapsules?
2.The microcapsule preparation method described in this article has a relatively low production efficiency. How can we improve the production efficiency? And, how can we remove the microcapsules from the mold without damaging them?
3.The data on drug release in the article indicates that there is a significant problem of premature release in this formulation. How can it be solved? Furthermore, this study lacks in vivo tests for drug release.
Author Response
Reviewer 1.
Comments 1: The microcapsules are oval in shape. How can we ensure the uniformity of the outer shell of the microcapsules?
Response 1: Thank you for pointing this out. The microcapsules are fabricated within a template, the shape of which is created using PDMS. To ensure uniformity of the outer shell, we employed a laser pyrolysis technique that converts PDMS into easily removable silicon carbide. This process involves sequential photothermal pyrolysis guided by a continuous-wave laser. The continuous-wave irradiation enables smoother and more controlled burning traces, which are subsequently imprinted onto the surface of the microcapsule shells. This templating approach ensures consistent geometry and shell uniformity across batches. To be precise, the laser parameters that provide sustained wave annealing are presented in Section 2.3, paragraphs 1 and 2.
Comments 2: The microcapsule preparation method described in this article has a relatively low production efficiency. How can we improve the production efficiency? And, how can we remove the microcapsules from the mold without damaging them?
Response 2: We appreciate this important observation. Indeed, the experiments described in the manuscript were performed at low throughput, as the primary aim of the study was to optimize polymer selection, concentration, and the drug loading approach within individual microcapsules. Nevertheless, the production efficiency can be significantly increased by scaling up the template size or by automating the process in a conveyor-like system. Such a system may include sequential stages of template immersion, drying, drug loading, polymer coating, and sealing, enabling continuous fabrication with higher yield.
Regarding the second part of your question, the removal of microcapsules from the mold was one of the challenges we encountered during the experimental phase. Through optimization of the polymer solution properties and behavior during solidification, we determined that a composite polymer system (PLA:PCL) offers advantages. It helps to mitigate thermal effects associated with PCL is used alone due to its lower melting point and preserves drug integrity, which may be compromised if PLA’s high melting point.
Two microcapsule release methods were evaluated.
Method 1 involves the use of a gelatin layer: a gelatin solution (typically 5–10% w/v, which gels upon cooling) is spread onto a flat surface such as glass slides or Petri dishes. The mold with microcapsules is placed face-down onto this surface and cooled (optionally in a freezer) to allow gelation. The gelatin layer adheres to the capsules, enabling their separation from the mold. Subsequently, the gelatin can be removed by gentle water washing and decantation.
Method 2 employs a thin polyvinylpyrrolidone (PVP) film as a receiving surface. The mold is gently pressed onto the film, transferring the microcapsules without exposing them to moisture. This dry transfer method preserves capsule integrity and allows storage in solid form.
These removal techniques and production scaling concepts are now mentioned in the revised manuscript (Section 2, Paragraph 2.3).
Comments 3: The data on drug release in the article indicates that there is a significant problem of premature release in this formulation. How can it be solved? Furthermore, this study lacks in vivo tests for drug release.
Response 3: Thank you for your thoughtful comment. We agree that premature release is a critical issue, especially in the context of controlled drug delivery. In our current formulation, early release was observed primarily due to the hydrophilic nature of the drug and potential imperfections in the sealing layer. This experimental phase was focused on screening suitable polymer compositions, and the PLA:PCL blend was selected to balance rigidity and flexibility. However, to further minimize premature release, several strategies are under consideration for future work:
- Improved sealing: Optimizing the sealing layer by introducing a solvent-welding step or applying a second outer coating..
- Multilayer encapsulation: Incorporating a multi-shell design using alternating layers of PLA and PCL to slow down water penetration.
Regarding the second point, we acknowledge the lack of in vivo data in this initial study. Our primary goal was to establish a proof-of-concept for the fabrication technique and to characterize the physical and release properties in vitro. Following successful stabilization of the release profile and extension of drug release duration, in vivo drug release studies will be planned to evaluate pharmacokinetics and therapeutic efficacy. This two-phase approach ensures that in vivo testing is conducted only after achieving adequate control over the release kinetics.
Reviewer 2 Report
Comments and Suggestions for Authors
Review of the manuscript “Microencapsulation of analgesics as an analogue form of medicine”.
Overall assessment:
The work presents an study on the microencapsulation of richlocaine using biodegradable polymers for the controlled release of water-soluble drugs. However, several critical methodological and characterisation issues emerged that limit the repeatability and efficacy assessment of the proposed system:
a) Incomplete description of release studies
The section on release studies is extremely concise and does not provide key information:
- The release medium used (type of solution, pH, volume, temperature) is not specified;
- The ratio of the amount of microparticles to the volume of the medium is not given;
- Details on release times (total duration, sampling intervals), stirring conditions and analytical methodology (calibration curve, wavelength, sensitivity) are missing;
- Any models of release kinetics or fitting of experimental data are not discussed.
The authors should include a detailed description of the experimental release protocol and discuss the results in terms of kinetics and the applied mathematical model.
b) Organic solvent residue control
Chloroform is used as a solvent during the manufacture of the microcapsules:
- No procedures for washing or purifying the microcapsules to remove chloroform residues are described.
- No specific analyses (e.g. GC-MS, HPLC) are described to verify the absence of residual solvents in the final microcapsules.
The authors should describe any washing/purification procedures and include analytical data confirming the absence of toxic solvent residues as a prerequisite for pharmaceutical application.
c) Stability of microcapsules in an aqueous environment
- No stability tests of the microcapsules in an aqueous or physiological environment are reported, nor are data on the structural integrity of the particles after immersion in solution.
The authors should perform stability tests in an aqueous environment (e.g. PBS, physiological pH, body temperature) to assess microcapsule strength and release consistency under simulated real-world conditions.
d) Quantities and experimental parameters
- Several steps lack quantitative details (masses, volumes, exact concentrations of polymer and drug used, times and temperatures of each step).
The authors should provide all the quantitative details necessary for the repeatability of the experiments.
e) Morphological and functional characterisation
- Morphological characterisation by SEM is adequately described, but quantitative data (size distribution, standard deviation, number of particles analysed) are lacking.
The authors should include the missing statistical data on the microcapsules.
Author Response
Reviewer 2.
Comments 1: a) Incomplete description of release studies.
The section on release studies is extremely concise and does not provide key information:
- The release medium used (type of solution, pH, volume, temperature) is not specified;
- The ratio of the amount of microparticles to the volume of the medium is not given;
- Details on release times (total duration, sampling intervals), stirring conditions and analytical methodology (calibration curve, wavelength, sensitivity) are missing;
- Any models of release kinetics or fitting of experimental data are not discussed.
The authors should include a detailed description of the experimental release protocol and discuss the results in terms of kinetics and the applied mathematical model.
Response 1: Thank you for your detailed and constructive feedback. We acknowledge that the original description of the release study lacked several critical methodological details, and we have now included the missing information in the revised manuscript (Materials and Methods section).The release was carried out in the physiological solution (0.9% NaCl in deionized water) at 37C. To ensure a uniform mixing, vials were subjected to a constant stirring on an orbital shaker at 300 RPM. 5 mсg of polymeric particles loaded with richlocaine were added into the 1 ml of saline solution. To carefully study the elution rate at it early moments, first samples was taken at 4, 8 and 24 hours respectively. The rest of data points were sampled each 24 hours until reaching the 72 hours mark. At each time point, the eppindorf vials were centrifuged to separate particles from the solution. The whole supernatant was then taken to the spectroscopic analysis and particles were resuspended within the fresh clean portion of the physiological solution. The spectroscopic study was carried out at 275 nm according to the calibration curve. 200 ul of supernatant was placed in the uv transparent 96 well-plate and absorption at 275 nm was evaluated. Using the equation of the calibration curve the absorbance value was converted to the richlocaine concentration.
Added to Discussion section in the paragraph describing Figure 8: It is common to use Korsmeyer-Peppas Model [10.5937/arhfarm72-40229] for describing the drug elution from the polymeric microstructers. It is clear from the release curve that the elution is non-linear and could be approximated by the power law to a certain accuracy. Korsmeyer-Peppas fitting reveals the elution rate constant value of 0.7522 and the power value of 0.092 which indicated purely diffusional drug release potentially caused by the highly pours wall of particles.
Comments 2: b) Organic solvent residue control
Chloroform is used as a solvent during the manufacture of the microcapsules:
- No procedures for washing or purifying the microcapsules to remove chloroform residues are described.
- No specific analyses (e.g. GC-MS, HPLC) are described to verify the absence of residual solvents in the final microcapsules.
The authors should describe any washing/purification procedures and include analytical data confirming the absence of toxic solvent residues as a prerequisite for pharmaceutical application.
Response 2: We thank the reviewer for raising this important issue regarding residual organic solvents. We fully acknowledge that chloroform is a toxic and non-pharmaceutically acceptable solvent and that its presence in the final product must be carefully controlled and minimized. To be avoided, if possible, there are some procedures to reduce it reaming quantities in final particles[10.1515/cdbme-2018-0136; 10.1515/cdbme-2016-0014]. According to toxicity data, relatively small dosage of particles (<10 ug/ml) exhibits minimal cytotoxic effect, but higher dosages clearly shows signs of toxicity, which can be related to the residual chloroform content as well as to several other factors. The main goal of this article was to highlight the novel dosage form of the richlocaine, suitable for prolonged therapy. In principle, in a case of pure PLA, choloroform can be avoided and replaced with “green” solvent like ethylacetate. The aforementioned reasons were incorporated into the discussion section in the paragraphs describing the choice of polymer concentration.
Comments 3: c) Stability of microcapsules in an aqueous environment
- No stability tests of the microcapsules in an aqueous or physiological environment are reported, nor are data on the structural integrity of the particles after immersion in solution.
The authors should perform stability tests in an aqueous environment (e.g. PBS, physiological pH, body temperature) to assess microcapsule strength and release consistency under simulated real-world conditions.
Response 3: Thank you for this important comment. Although, the particles stability is the important factor for the long term drug elution (weeks of mounts) experimental data confirms that described dosage form of richlocaine is suitable for sustained dosage within the first 24 hours. At such timescales both PLA and PCL are highly stable [10.1016/j.polymdegradstab.2016.03.037] and changes of particles due to hydrolysis and enzymatic activities can be neglected.
Comments 4: d) Quantities and experimental parameters
- Several steps lack quantitative details (masses, volumes, exact concentrations of polymer and drug used, times and temperatures of each step).
The authors should provide all the quantitative details necessary for the repeatability of the experiments.
Response 4: Thank you for your valuable comment. We fully agree that providing precise quantitative parameters is essential for ensuring the reproducibility of experimental procedures. In response to this, we have revised and expanded Sections 2.1, 2.2, and 2.3 of the Materials and Methods to include detailed information. It is important to note that exact masses and volumes of polymer solutions were prepared individually for each experiment depending on scale, and were therefore not fixed. However, the percentage compositions and concentration ratios were kept consistent and are clearly specified in the revised manuscript.
Comments 5: e) Morphological and functional characterization
- Morphological characterisation by SEM is adequately described, but quantitative data (size distribution, standard deviation, number of particles analysed) are lacking.
The authors should include the missing statistical data on the microcapsules.
Response 5: Thank you for this comment. As the microcapsules were fabricated using a template-based method, the resulting particles exhibit minimal size variation, since their dimensions are predefined by the mold geometry. Due to this controlled manufacturing approach, quantitative size distribution analysis was not considered critical at this stage. Furthermore, the short incubation period of up to 72 hours is insufficient to cause significant structural degradation or dimensional changes, although minor surface modifications may occur as a result of rihlocaine diffusion.
Reviewer 3 Report
Comments and Suggestions for Authors
The preparation of drug-loaded microcapsules to achieve prolonged or controlled drug’s form is an important technological problem. After reading this manucript I have several doubts and comments.
• 2.1. Polymer composite preparing
What type of the apparatus was used? – name, a manufacturer
What operating process conditions were used?
• 2.2. Fabrication of Sealed Free-Standing Polymer Films with Microchambers Array Containing
- no detailed description of preparation
• 2.4. Characterization Technique The morphology and size distribution were assessed using scanning electron microscopy and optical microscopy. Additionally, this drug delivery system was evaluated by determining capsule capacity and drug release.
- Lack the chemical analytical methods to evaluate the content drug and drug efficiency microcapsules.
• 3. Results and Discussion In light of the SEM images acquired, an assessment of the release kinetics was conducted for a cohort of microcapsules, characterized by a shell composition comprising
The liberation kinetics of richlocaine from the microcapsules were scrutinized within a saline solution using a thermostatic shaker. Upon careful examination of the release profile, it becomes evident that a predominant fraction of the drug is liberated within the initial 8-hour period, with sustained release plateauing thereafter, exhibiting minimal changes even at the 24-hour mark (Figure 8).
- The release test should be carried out according to the pharmacopoeial requirements (for example European Pharmacopoeia or another) using appropriate a dissolution medium and an apparatus. Kinetic data of release can be calculated according the mathematical models reported in literature for example Higuchi or another.
Figure 8 is not show the profile of prolonged or controlled release. Lack results the periods of time release drug 20%, 50% and 80%
Author Response
Reviewer 3.
Comments 1: Polymer composite preparing. What type of the apparatus was used? – name, a manufacturer. What operating process conditions were used?
Response 1: Thank you for your observation. The polymer composite preparation described in Section 2.1 was carried out using a laboratory magnetic stirrer — IKA C-MAG 7 digital manufactured by IKA (Germany). The polymers were dissolved in dichloromethane in tightly sealed borosilicate glassware to prevent solvent evaporation. To fabricate a composite polymer, individual solutions were initially prepared separately due to differing solubility temperatures. Solutions with concentrations of 1%, 1.5%, and 3% were formulated. Following preparation, the solutions were combined in a volumetric ratio of 30:70 based on mass, ensuring uniform distribution within the composite matrix.
Operating conditions of composite polymer solution were as follows:
- Stirring time: 60 minutes
- Stirring speed: 300 rpm
- Ambient temperature: 22–24 °C
These conditions were selected to ensure homogeneous dissolution of high-molecular-weight polymers (PLA and PCL), while minimizing solvent loss and maintaining viscosity stability.
We have added the name and manufacturer of the apparatus, as well as the operating parameters, in the revised manuscript (Section 2.1, final paragraph).
Comments 2: Fabrication of Sealed Free-Standing Polymer Films with Microchambers Array Containing - no detailed description of preparation
Response 2: We agree that the initial version of the manuscript lacked sufficient detail in the section “Fabrication of Sealed Free-Standing Polymer Films with Microchambers Array.” We have now added a more comprehensive description of the preparation procedure.
Specifically, the pre-fabricated mold — containing predefined microchamber geometry — was immersed into a pre-prepared polymer solution for 5 seconds and then removed and suspended vertically to dry under ambient conditions. During drying, the solvent gradually evaporates, resulting in the formation of a thin polymer film conforming to the mold's topography. This initial film serves as the outer shell of the future microcapsules.
After complete drying, the active pharmaceutical ingredient was loaded into the microchambers, followed by application of a pre-fabricated flat polymer film as a sealing layer. This top film was gently laminated over the filled surface to enclose the contents and form sealed microcapsules. The combined structure was allowed to settle, ensuring proper adhesion and sealing.
These steps have now been described in the revised manuscript (Section 2.2), as requested.
Comments 3: Characterization Technique The morphology and size distribution were assessed using scanning electron microscopy and optical microscopy. Additionally, this drug delivery system was evaluated by determining capsule capacity and drug release. Lack the chemical analytical methods to evaluate the content drug and drug efficiency microcapsules.
Response 3: Thank you for this important comment. We agree that a more detailed description of the chemical analytical methods is necessary to evaluate the drug content and efficiency of microcapsules.
In the present study, Fourier Transform Infrared Spectroscopy (FTIR) was used to confirm the qualitative presence of the active compound (rihlocaine) within the polymer matrix in the Results and Discussion section, Figure 7.
We acknowledge, however, that quantitative chemical analysis, such as UV–Vis spectrophotometry, HPLC, or LC–MS, is typically required to accurately determine drug loading, encapsulation efficiency, and release profiles. In our case, the primary goal of the current work was to establish the feasibility of fabrication and demonstrate physical characteristics and qualitative drug retention.
We are planning a follow-up study that will improve the microcapsule process and properties based on several comments, and further test and utilize UV–Vis and HPLC methods to quantify encapsulated and used to quantify encapsulated drug amounts and release kinetics over time. This perspective has been added to the Discussion section.
Comments 4: Results and Discussion In light of the SEM images acquired, an assessment of the release kinetics was conducted for a cohort of microcapsules, characterized by a shell composition comprising. The liberation kinetics of richlocaine from the microcapsules were scrutinized within a saline solution using a thermostatic shaker. Upon careful examination of the release profile, it becomes evident that a predominant fraction of the drug is liberated within the initial 8-hour period, with sustained release plateauing thereafter, exhibiting minimal changes even at the 24-hour mark (Figure 8). The release test should be carried out according to the pharmacopoeial requirements (for example European Pharmacopoeia or another) using appropriate a dissolution medium and an apparatus. Kinetic data of release can be calculated according the mathematical models reported in literature for example Higuchi or another. Figure 8 is not show the profile of prolonged or controlled release. Lack results the periods of time release drug 20%, 50% and 80%
Response 4: Thank you for your detailed and valuable comment. We agree that the release test, as presented, does not yet meet the full rigor of pharmacopoeial standards. In this study, our primary aim was to investigate the influence of polymer composition and structural parameters on microcapsule formation and drug entrapment, using a simplified in vitro model to identify trends in release behavior.
Additionally, we do not exclude the possibility that the observed high initial release may be partially attributed to structural defects in the microcapsules, such as incomplete sealing or thin regions in the outer polymer layer, leading to premature leakage. Moreover, the rate of drug release may also be influenced by the biodegradation profiles of the polymers used: poly(lactic acid) (PLA) typically degrades in vivo over a period of 6 to 12 months, while poly(ε-caprolactone) (PCL) can take 12 to 36 months, depending on molecular weight, crystallinity, and environmental conditions. These factors should be further investigated in future work using long-term release studies and stability assessments.
We acknowledge that a validated dissolution medium and apparatus, such as USP Apparatus I or II, should be employed to ensure reproducibility and regulatory relevance. Unfortunately, at this stage of the research, access to the required apparatus is not available. However, the authors recognize the importance of such analyses and plan to incorporate them in future studies once the necessary equipment becomes accessible. Furthermore, in the next phases of our research, we plan to apply mathematical models of drug release kinetics, including Higuchi, Korsmeyer–Peppas, and zero/first-order models, to better interpret and predict the release mechanism.
Round 2
Reviewer 1 Report
Comments and Suggestions for Authors
The author has replied to all the questions.
Reviewer 2 Report
Comments and Suggestions for Authors
I have carefully reviewed the authors’ responses to my previous comments and the revised version of the manuscript. I appreciate the thorough and thoughtful way in which the authors have addressed all the points raised. The manuscript has significantly improved and now meets the scientific and editorial standards of the journal.
I recommend acceptance in its current form.
Reviewer 3 Report
Comments and Suggestions for Authors
I accept the author's explanations.